# Mitochondrial Dysfunction in Systemic Lupus Erythematosus with a Focus on Lupus Nephritis

**DOI:** 10.3390/ijms25116162

**Published:** 2024-06-03

**Authors:** Matthieu Halfon, Aurel T. Tankeu, Camillo Ribi

**Affiliations:** 1Transplantation Center, Lausanne University Hospital, Rue du Bugnon 44, CH-1010 Lausanne, Switzerland; aurel.tankeu-tiakouang@chuv.ch; 2Division of Immunology and Allergy, Lausanne University Hospital, CH-1010 Lausanne, Switzerland; camillo.ribi@chuv.ch

**Keywords:** lupus, kidney, mitochondria, mitochondrial DNA, mitophagy, interferon, lupus nephritis

## Abstract

Systemic lupus erythematosus (SLE) is an autoimmune disease affecting mostly women of child-bearing age. Immune dysfunction in SLE results from disrupted apoptosis which lead to an unregulated interferon (IFN) stimulation and the production of autoantibodies, leading to immune complex formation, complement activation, and organ damage. Lupus nephritis (LN) is a common and severe complication of SLE, impacting approximately 30% to 40% of SLE patients. Recent studies have demonstrated an alteration in mitochondrial homeostasis in SLE patients. Mitochondrial dysfunction contributes significantly to SLE pathogenesis by enhancing type 1 IFN production through various pathways involving neutrophils, platelets, and T cells. Defective mitophagy, the process of clearing damaged mitochondria, exacerbates this cycle, leading to increased immune dysregulation. In this review, we aim to detail the physiopathological link between mitochondrial dysfunction and disease activity in SLE. Additionally, we will explore the potential role of mitochondria as biomarkers and therapeutic targets in SLE, with a specific focus on LN. In LN, mitochondrial abnormalities are observed in renal cells, correlating with disease progression and renal fibrosis. Studies exploring cell-free mitochondrial DNA as a biomarker in SLE and LN have shown promising but preliminary results, necessitating further validation and standardization. Therapeutically targeting mitochondrial dysfunction in SLE, using drugs like metformin or mTOR inhibitors, shows potential in modulating immune responses and improving clinical outcomes. The interplay between mitochondria, immune dysregulation, and renal involvement in SLE and LN underscores the need for comprehensive research and innovative therapeutic strategies. Understanding mitochondrial dynamics and their impact on immune responses offers promising avenues for developing personalized treatments and non-invasive biomarkers, ultimately improving outcomes for LN patients.

## 1. Introduction

Systemic lupus erythematosus (SLE) stands out as an archetypal autoimmune disease, affecting predominantly women of child-bearing age, with an incidence ranging from 1.4 to 10 cases per 100,000 people annually. Its prevalence varies significantly among different ethnic groups, with the lowest rates observed in Asian descents and the highest among Afro-Caribbean populations [1]. In Europe, the estimated prevalence ranges from 40 to 80 people per 100,000 individuals [2]. The underlying autoimmune dysregulation is due to a combination of genetic and environmental factors [3]. Key genes primarily involved in SLE pathogenesis include elements within the toll-like receptor (TLR), type-I interferon (IFN-I), consisting of IFN-alpha and IFN-Beta, and complement pathways [3]. It is noteworthy that elements from both the innate and adaptive immune systems contribute to the immune dysfunction in SLE. A defining characteristic of this condition is the disruption of normal apoptosis mechanisms, leading to inappropriate cellular death and the impaired clearance of cellular debris. This leads to the accumulation of cellular remnants, fostering a loss of immune tolerance and the subsequent generation of autoantibodies. These autoantibodies form immune complexes that initiate complement activation, culminating in organ damage [4].

Given their susceptibility to immune complexes, the kidneys are frequently affected in SLE. Lupus nephritis (LN) serves as the initial presentation of SLE in approximately 16% of patients and appears in SLE with an overall prevalence ranging from 30% to 40% [5]. Furthermore, a significant proportion of SLE patients (20% to 50%) are at risk of developing LN within the first year following their initial diagnosis of SLE [5]. Despite recent breakthroughs in the development of new therapeutics, it is crucial to acknowledge that the remission rate for LN remains low, from 30% to 70% [6,7,8,9,10,11]. This fact holds significant importance because achieving remission in LN is intricately linked to the risk of progressing to end-stage kidney disease (ESKD). Even with optimal treatment, a notable subset of patients (5% to 20%) will ultimately develop ESKD which is associated with a substantial burden [12].

Recent experimental studies, conducted on animal models and human subjects, have provided compelling evidence of altered immune cell metabolism contributing to SLE pathogenesis. Indeed, immune cells in SLE exhibit heightened metabolic demands [13,14,15]. Given that mitochondria are the central hub for cellular metabolism, it is postulated that these organelles play a pivotal role in the development and amplification of the immune dysfunction in SLE. Defects in mitochondrial pathways, such as compromised mitophagy mechanisms and impaired mitochondrial DNA (mtDNA), have been recently considered key players in the pathogenesis of lupus. Mitochondrial dysfunctions were linked to the heightened production of IFN-I, a critical cytokine in SLE. In lupus-prone mice, the inhibition of glycolysis has been shown to result in a significant reduction in the production of autoantibodies and a decrease in T cell activation [16,17]. On the other side, mitochondrial dysfunction is seen in various kidney diseases, in the setting of acute kidney injury but also in chronic kidney disease [18,19,20].

In this review, our primary emphasis is on elucidating the interplay between mitochondria and the activation/amplification of SLE, with focus on LN. Additionally, we explore the usefulness of mitochondrial products as biomarkers and as targets for therapeutic interventions in SLE.

## 2. Mitochondria Anatomy, Role, and Functions

Mitochondria are small intracellular compartments with unique anatomical and physiological characteristics, ranging in size from 0.5 to 3 μm. Their main role is energy production in the cell through an electron transport chain (ETC) and oxidative phosphorylation (OXPHOS) [21]. Mitochondria exist in almost all eukaryotic cells and are thought to have bacterial origin [22]. Indeed, similarly to bacteria, mitochondria are surrounded by a double-stranded membrane, possess their own genome known as mtDNA, and rely on specific ribosomes that are vulnerable to antibiotics [23]. The double membrane consists of an outer layer (OMM) and an inner layer separated by the intermembrane space. The inner mitochondrial membrane (IMM) forms numerous folds called cristae, which extend into the free space demarcated by the double membrane, called the mitochondrial matrix. The location of the mtDNA close to the IMM and the cristae, together with the lack of introns and histones, makes it highly susceptible to intrinsic aggression, mostly due to reactive oxygen species (ROS) generated by mitochondria complexes during OXPHOS. Given all the above and due to a less effective DNA repair system, mtDNA has a higher mutation rate (10–20 times higher) than for nuclear DNA, and alterations in its sequence (mutations, insertion, and deletions) and structure (rearrangements and breaks) are frequent. In humans, the number of copies of mtDNA per mitochondria varies from 5 to 10 [24]. The multiple copies of mtDNA within a cell and its high susceptibility to alterations favors the coexistence of several mtDNA populations (wild type and mutated mtDNA) in the same cell. This phenomenon is called heteroplasmy, as opposed to homoplasmy which is the existence of unique types of mtDNA within the cell (less frequent). The proportion of mutated mtDNA in relation to total mtDNA determines the heteroplasmy rate [25].

Similarly, to mtDNA, mitochondrial macromolecules (lipids and proteins) can be damaged by ROS, leading to mitochondrial dysfunction. Healthy mitochondria are critical for cell survival, while dysfunctional mitochondria promote apoptosis through Ca2+ and cytochrome *c* release to the cytosol [26]. The cellular population of mitochondria (mitochondria mass and quality) is regulated through fine-tuned processes, including the generation of new mitochondria through biogenesis (new mitochondria formation), the fusion or fission of existing mitochondria, and the degradation of damaged mitochondria by mitophagy [27] (Figure 1).

In response to specific needs or stresses such as exercise, cold, or fasting, mitochondria biogenesis is activated through its master regulator: the peroxisome proliferator-activated receptor-γ coactivator-1α (PGC-1α), which activates mtDNA transcription and increases in the expression of mitochondrial transcription factor A (TFAM), the final effector of mtDNA transcription and replication [27,28]. Mitochondrial fission is the division of a mitochondrion into two distinct mitochondria. This process plays a role in inheritance and mitochondrial partitioning during cell division, apoptosis, and mitophagy [29]. Mitochondrial fission in mammals is coordinated by a GTPase dynamin-related protein 1 (DRP1), which act in conjunction with mitochondrial fission 1 (FIS1), and mitochondrial fission factor (MFF). DRP1, a cytosolic protein composed of four domains, is recruited by its adaptors present on the OMM, (MFF and FIS1), where it undergoes the formation of an oligomeric ring structure around the mitochondrion, further constricting it. Then, the GTP hydrolysis of DRP1 completes the mitochondrion cleavage process.

Defective or damaged mitochondria that lost their membrane potential are inclined to release high amounts of Ca^2+^ and cytochrome *c* to the cytosol, which promotes cell apoptosis [26]. To prevent this, dysfunctional mitochondria are degraded through a specific autophagy-dependent process known as mitophagy.

Mitophagy can be classified into receptor-dependent (also called Parkin-independent) mitophagy and ubiquitin-dependent (or Parkin-dependent) mitophagy [26,30]. In mammals, receptor-dependent mitophagy is a three-step process starting with the activation of receptors located on the OMM (FUNDC1, NIX/BNIP3L, BNIP3, or Bcl2L13), followed by the binding of the autophagosome marker LC3, which initiates the development of the phagophore membrane and forms the autophagosome, and finally, the fusion of the autophagosome with the lysosome for cargo degradation [26].

Ubiquitin is a protein involved in protein and organelle degradation through ubiquitination, which is the conjugation of ubiquitin to proteins mediated by three enzymes: ubiquitin-activating enzymes (E1), ubiquitin-conjugating enzymes (E2), and ubiquitin-protein ligases (E3) [31]. The ubiquitin proteasome system (UPS) is a proteasome-mediated degradation machinery that removes proteins from various cellular compartments. Autophagy and UPS represent the main intracellular proteolysis machineries that enforce protein and organelle quality control in the cell [30].

Both systems utilize ubiquitin signaling to tag their targets, thus cooperating in the elimination of damaged and dysfunctional mitochondria. By tagging substrates to be degraded by autophagy or by UPS, ubiquitin is therefore a common signal for protein or organelle degradation [26]. One of the most well-characterized ubiquitin-dependent autophagy is the mitophagy pathway; it involves the PTEN-induced putative kinase 1 (PINK1) and the E3 ubiquitin-protein ligase Parkin [32]. Here, the initial step is the accumulation of PINK1 at the OMM following the loss of mitochondria membrane potential. PINK1 accumulation phosphorylates and activates Parkin that is recruited to the OMM where it ubiquitinates OMM proteins using its E3 ligase activity. Ubiquitinylated OMM proteins are recognized by specific adaptors, allowing the engulfment of mitochondria and formation of autophagosomes with the recruitment of LC3 via an LC3-interacting region (LIR) motif [32]. The fusion and degradation of autophagosomes with lysosomes are then consistent with receptor-mediated mitophagy [26]. Mitophagy is highly responsive to the dynamics of endogenous metabolites, including iron-, calcium-, glycolysis-TCA-, NAD+ -, amino acids-, fatty acids-, and cAMP-associated metabolites [33]. For instance, the disruption of mitochondrial membrane potential is a potent mitophagy activator, but SIRT3, a mitochondrial sirtuin and NAD + metabolic sensor, has been shown to restore the proton gradient, playing a role in maintaining mitochondrial membrane potential in response to mitochondrial stress, therefore reducing mitophagy [34,35]. Mitophagy can start with fission, which help in the fragmentation of mitochondria before their degradation. It has been shown that the mitochondrial recruitment of Drp1 is a crucial step to initiate mitophagy [36]. HRES-1/Rab4 promotes the lysosomal degradation of Drp1; therefore, HRES-1/Rab4 induces the accumulation of mitochondria by inhibiting mitophagy [37].

## 3. Mitochondrial Dysfunction: A Trigger and Amplifier of Type I Interferon

Type I IFNs represent a group of inflammatory cytokines that play a pivotal role in the body’s defense against viral infections, primarily by the activation of TLRs in response to viral particles. Through pleiotropic mechanisms, IFN-I activates both T and B-cells [38]. For decades, SLE has been recognized as a quintessential interferonopathy. This designation stems from the robust expression of IFN-I seen in most SLE patients. The high IFN-I signature in SLE has long been attributed to activated plasmacytoid dendritic cells (pDCs). These pDCs are implicated in the proliferation and survival of autoreactive lymphocytes, which significantly contribute to the production of autoantibodies and the formation of immune complexes. However, recent findings in SLE challenge the notion that pDCs are the primary source of IFN [39]. Neutrophils undergoing NETosis have emerged as significant IFN producers via the stimulator of interferon gene (STING) pathway, notably contributing to the IFN gene signature observed in the kidneys of patients suffering from LN [40]. Also, a novel concept involves “local” IFN release by non-hematopoietic cells such as epithelial or proximal tubular cells in the kidney, releasing IFN [41,42]. This local, non-circulating IFN production may contribute to the diverse phenotypes observed in SLE. Of note, GDF-15, a cytokine released locally by various organs in cases of mitochondrial stress, is known to regulate IFN production [43].

A substantial proportion of individuals with SLE exhibit an upregulation of type I IFN-regulated genes, collectively referred to as the “IFN-I signature,” both in their blood and affected tissues. IFN-I signatures correlate with SLE activity. They have been used in clinical trials to stratify patient groups for treatments targeting the IFN-I pathway, such as anifrolumab, an IgG1κ monoclonal antibody (mAb) blocking the IFN-I receptor subunit 1. Anifrolumab was recently approved for the treatment of SLE, which underscores the value of targeting IFN-I pathways in SLE. Other approaches, such as targeting pDCs as potential source of INF-I in SLE, are currently under investigation, such as the mAB litifilimab, which targets the blood dendritic cell antigen 2 (BDCA2) [44,45]. A summary of the current drugs used to treat LN is provided in Table 1.

IFN-I production is amplified by dysregulated mitochondrial homeostasis through various mechanisms. First and foremost, mtDNA itself acts as a danger-associated molecular pattern (DAMP) due to its unmethylated CpG sequence’s structural resemblance to bacterial DNA. This structural similarity to bacterial DNA enables TLR9 activation, ultimately leading to the release of IFN-I [46]. In SLE, high levels of circulating oxidized mtDNA (oxmtDNA) have been identified within circulating neutrophils. What is particularly intriguing is that oxmtDNA has been found to possess a heightened propensity for internalization by pDCs, thus potentially contributing to the IFN-I signature seen in SLE [47,48]. Under normal physiological conditions, oxmtDNA undergoes a process of degradation within the lysosome. The endocytosis of oxmtDNA into the lysosome is facilitated by its dissociation from TFAM. Caielli and colleagues have demonstrated that autoantibodies targeting ribonucleotide proteins (anti-RNP) may obstruct TFAM phosphorylation, which in turn prevents oxmtDNA dissociation, rendering it resistant to degradation. This disruption in the natural degradation of oxmtDNA degradation may contribute to the vicious circle characterizing immune dysregulation in SLE [47] (Figure 2).

Another contributor to the immune dysfunction in SLE is the formation of extracellular traps, first described in neutrophils and termed “NETosis” [49]. Extracellular trap formation (ETF) is primarily used as a defense mechanism against pathogens. Cells capable of ETF extrude proteins and DNA to form a biological “web”, intended to trap microorganisms. In neutrophils, NETosis was thought to be ineluctably associated with cell death (suicidal or lytic NETosis). Recently, another type of NETosis was reported, where cell functions are preserved (vital NETosis) [50]. It has been observed that IFN-I can trigger NETosis in SLE. On the other hand, oxmtDNA is a major component of extruded cell material during NETosis [48]. ETF thus may constitute another amplifying loop, where oxmtDNA enhances IFN-I production, while IFN-I-regulated genes promote ETF with the release of additional oxmtDNA. Finally, mitochondrial reactive oxygen (mtROS) species also act as DAMPs and may enhance IFN-I production by provoking the oligomerization of mitochondrial antiviral stimulator (MAVS) and ETF [51].

Neutrophils are not the only cells capable of ETF and sources of extracellular mitochondrial material. It was recently shown that platelets activated by immune complexes also release mitochondrial material [52]. Furthermore, recent findings have revealed that mtDNA that has leaked into the cytosole may also serve as a proinflammatory stimulus, boosting the cell’s IFN-I expression [53]. The leakage of mtDNA to the cytosol occurs through pores in the OMM and activates the BAX/BAK pathways and subsequently caspase-9, thereby initiating the apoptosis cascade [54]. An alternative mechanism for the cytosolic release of mtDNA has also been described by Kim and colleagues [55]. They describe that the oligomerization of the voltage-dependent anion-selective channel 1 (VDAC1) is responsible for allowing mtDNA to leak into the cytosol, particularly under conditions of oxidative stress observed in lupus-prone mouse models.

Mitophagy prevents the release of mitochondrial materials into the extracellular environment, which is pivotal in maintaining cellular homeostasis. In SLE, various findings hint at defective mitophagy. Impaired mitophagy likely contributes to the release of mitochondrial components to the cytosol and the extracellular compartment, thus amplifying SLE immune dysregulation and, in particular, heightened IFN-I response.

CD4+ T cells in individuals with SLE exhibit an elevated mitochondrial mass, primarily attributed to a dysfunction in mitophagy [56]. Among the mechanisms contributing to this impaired mitophagy in CD4+ T cells is the overexpression of HRES-1/Rab4, a GTPase enzyme that promotes the degradation of Drp1 [14]. CD8+ T cells also have mitochondrial impairment. Elevated CD38 expression in CD8 + l T cells leads to decreased mitochondrial endocytosis by inhibiting sirtuin protein activity. Furthermore, it reduces V-ATPase activity, hindering lysosomal acidification due to diminished NAD+ levels. Consequently, the inability to internalize or degrade mitochondria within lysosomes contributes to the accumulation of dysfunctional mitochondria, increased mitochondrial mass, and the disruption of cellular function [57]. This accumulation further triggers the release of DAMPs like mtDNA and mitochondrial ROS, subsequently activating the STING pathway and inducing IFN production. Importantly, it is worth noting that IFN itself can enhance the expression of CD38 [56]. Mitophagy dysfunction is not confined to immune cells in SLE. Caielli and colleagues have recently revealed that a unique population of red blood cells, distinct to SLE patients, retains their mitochondria. This unusual phenomenon results from a disruption in the metabolic pathway responsible for transitioning between glycolysis and oxidative phosphorylation in mitochondria. Indeed, this transition is essential for activating the ubiquitin–proteasome system, which plays a pivotal role in mitophagy regulation [58]. Furthermore, in some SLE patients, there are antibodies that can bind to and opsonize red blood cells. When these opsonized red blood cells are encountered by myeloid cells, the mitochondria within them serve as DAMPs and are a potential source of IFN release [58].

## 4. Mitochondrial Dysfunction in Lupus Nephritis

Mitochondrial damage is associated with the progressive decline of renal function in chronic kidney diseases (CKDs) related to various conditions and LN in particular. Recent research by Luan and colleagues, employing electron microscopy in kidney biopsies, has unveiled several mitochondrial abnormalities in LN patients, including fission occurring within podocytes [59]. Furthermore, a connection between mtDNA and ETF in the kidney was established by Wang et al., who demonstrated the deposition of mtDNA within NETs found in the kidney biopsies of patients experiencing active LN [60].

Specific expression patterns in mitochondrial genes are seen in LN, progressing to renal fibrosis. Particularly, circular RNA originating from the mitochondrial gene MTND5 exhibits downregulation in LN-afflicted renal cells. Circular RNAs, a recently identified class of non-coding RNA fragments, are recognized for their role in regulating protein expression by modulating microRNAs, thereby influencing either the inhibition or activation of their function. In the case of circular MTND5, it normally downregulates microRNA6812, which is known to facilitate the activation of genes associated with fibrosis [59]. Moreover, a recent study by Tian et al. closely linked defective mitophagy to podocyte injuries, suggesting it could play a role in proteinuria in LN. First, they demonstrated that the podocyte expression of nestin was inversely correlated with proteinuria in LN. Furthermore, the expression of nestin was also linked to the expression of nephrin, a key protein of the glomerular basal membrane. Finally, they showed that nestin could regulate nephrin by inducing mitophagy through the PINK1 pathway [61].

Expanding on these observations, numerous authors have delved into the exploration of mitochondrial DNA as a surrogate marker for mitochondrial dysfunction in the context of LN. Notably, there has been a surge of enthusiasm surrounding the assessment of cell-free mitochondrial DNA as a potential biomarker.

In a cohort study comparing 80 SLE patients to 43 healthy controls, Hui-tin Lee and colleagues discovered that compared to the healthy controls, SLE patients exhibited higher levels of relative cell-free nuclear DNA (CFnDNA) and lower levels of relative cell-free mitochondrial DNA (CFmtDNA). Within SLE, individuals with active disease, indicated by an SLEDAI > 8, demonstrated even lower levels of CFmtDNA. Patients with LN showed a trend toward lower CFmtDNA compared to non-renal SLE. To explain this difference between mtDNA and nuclear DNA, the authors suggest that SLE patients were undergoing more vital NETosis than lytic NETosis, the former releasing a higher amount of nuclear DNA than mtDNA into the circulation. Furthermore, it is important to note that during NETosis, mitochondrial DNA becomes entrapped in the NET and is not freely found in the plasma [62]. The observation of lower CFmtDNA in LN is interesting, considering the pivotal role NETosis plays in this pathology [63,64,65]. Others have found a positive association between the total-mtDNA-to-CFmtDNA ratio and renal damage in SLE (eGFR < 60 mL/min). In this study, a lower CFmtDNA copy number was associated with proteinuria. The cell-free DNA profile failed, however, to discriminate patients with proliferative LN (Class III A and Class IV A) [66]. Another study conducted by Hui-Tin and colleagues delves into the “qualitative” dimension of mtDNA in SLE [67]. Their focus centers on the heteroplasmy of the D310 region, specifically examining variations in the number of cysteine and thymidine (D310 polymorphism) in the mtDNA of leukocytes. This particular mutation is the most prevalent, and its presence in this region raises suspicions of potential interference with the replication of mitochondrial DNA [68,69]. Their findings revealed that SLE patients exhibited a higher degree of D310 heteroplasmy compared to a control group. Moreover, there was a noticeable trend towards increased heteroplasmy with the progression of disease activity in the SLE group, as measured by the SLEDAl, and this corresponded to lower mtDNA copy numbers and a reduced expression of mitochondrial RNA genes. Moreover, patients with renal involvement demonstrated an even greater degree of D310 heteroplasmy [67].

In another study, Wang et al. demonstrated the presence of antibodies against mtDNA in 41% of patients within their cohort. Furthermore, these antibodies exhibited higher positivity in patients classified under Class III or IV A compared to Class III or IV C, although it is crucial to note the limited number of patients within Class III or IV C (N = 3) [60].

Collectively, these studies underscore the potential of mtDNA as a marker in SLE, particularly in the context of LN, offering promise for diagnostic applications. However, a cautious approach is warranted in interpreting these findings due to the existing heterogeneity in mtDNA quantification methods. Presently, these results lack validation in large cohorts and lack a robust direct comparison with the histology of renal biopsies from SLE patients, underscoring the necessity for circumspection when drawing definitive conclusions and assessing clinical utility. Furthermore, considering the intricate interplay between mitochondrial damage and interferon production, a comparison with classic interferon markers, such as the “interferon gene signature”, could provide valuable insights for a comprehensive assessment.

## 5. Mitochondria as Therapeutic Targets in SLE

Given the close relationship between mitochondria and interferon production in SLE, mitochondria stand out as a potential new target for therapy. Remarkably, there are currently affordable and readily available drugs that specifically target mitochondrial dysfunction, such as metformin or mTOR inhibitors. These drugs hold promise in modulating mitochondrial function and could potentially offer therapeutic benefits in managing SLE by targeting this critical cellular pathway.

Indeed, Lai et al. conducted a study on patients experiencing persistent lupus activity, examining the effects of sirolimus. The rationale behind this investigation was the known impact of mTOR inhibitors, such as sirolimus, on mitophagy. This effect is believed to occur due to their interaction with HRES-1/Rab4 expression [14]. In Lai’s study, it was found that 55% of patients treated with sirolimus experienced a reduction in SLEDAI and BILAG scores during the study period. However, it is crucial to note that over 25% of patients were excluded from the final analysis due to intolerance and non-compliance, potentially introducing bias into the results. Furthermore, the exclusion of patients with proteinuria raises questions about the potential role of sirolimus in LN since mTOR inhibitors have been associated with inducing proteinuria and focal segmental glomerulosclerosis. This limitation suggests a restricted scope regarding its application in LN management [70]. However, Wang et al. recently studied the efficacity of UMI-77, a molecule that restores mitophagy, in LN [71]. UMI-77 potentiates mitophagy by blocking the interaction between MCL-1 and Bax/Bak pathways, allowing MCL-1 to interact with LC3 and induce mitophagy. In their study on lupus-prone mice, the authors demonstrated that the kidneys of lupus mice treated with UMI-77 exhibited less glomerular inflammation, a reduced infiltration of inflammatory cells, fewer crescent formations, less fibrosis, and fewer immune complex deposits compared to the control. Interestingly, the effect of UMI-77 was not directly on the mitochondria of podocytes or tubular cells but rather on plasmacytoid dendritic cells, restoring their mitochondrial homeostasis and thereby contributing to reducing T cell infiltration in the kidney. Their study not only provides proof of concept for the role of mitophagy in LN but also serves as a salient milestone that will pave the way for further research on mitophagy as a therapeutic target of SLE [71]. In animal model studies, drugs known to enhance mitochondrial metabolism, such as idebenone (an analogue of coenzyme Q10) or mitoQ, have demonstrated improvements in renal histology [51,72]. These improvements were evident in reduced fibrosis, diminishing immune deposits on glomeruli. In vivo, those molecules also diminish NET formation and markers of IFN. However, most studies focusing on targeting mitochondria in LN remain in the preliminary stages, suggesting a long and extensive journey before their translation into clinical utility. A summary of the mitochondrial-targeting drugs in development for SLE is provided in Table 2.

## 6. Conclusions

LN stands as a complex disease, currently hindered by a limited array of new treatments. Its characterization is marked by the absence of non-invasive biomarkers, which are crucial for diagnosing active disease or assessing treatment response. The emerging role of mitochondria as a pivotal nexus for various essential mechanisms in the pathophysiology of LN represents an area yet to be fully explored and understood. Research in this domain holds the promise of eventually paving the way for more personalized and precisely tailored therapies in LN.

## Figures and Tables

**Figure 1 ijms-25-06162-f001:**
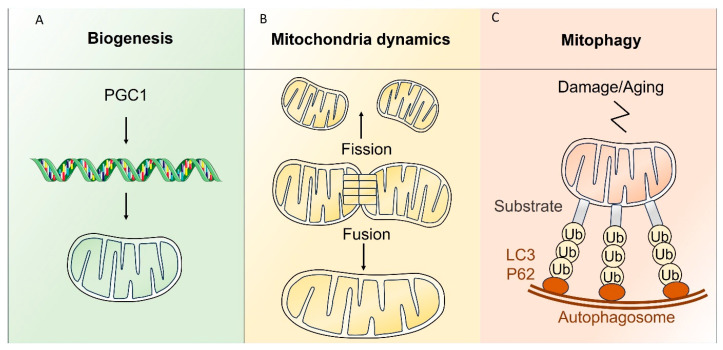
Quality control mechanism in mitochondria. Figure legend: (**A**) Biogenesis: activation of peroxisome proliferator-activated receptor-γ coactivator-1α (PGC1) due to stress factors, then activation of mt DNA transcription leading to formation of new mitochondria. (**B**) Mitochondria dynamic: fission of a mitochondrion resulting in two separate mitochondria. Fusion: Two mitochondria fusion at the outer and inner membrane interfaces. This process allows for exchange of mtDNA, proteins, or metabolites and improves overall mitochondria respiratory function and efficiency. (**C**) Mitophagy: degradation of the mitochondria with the ubiquitin pathway into the phagosome.

**Figure 2 ijms-25-06162-f002:**
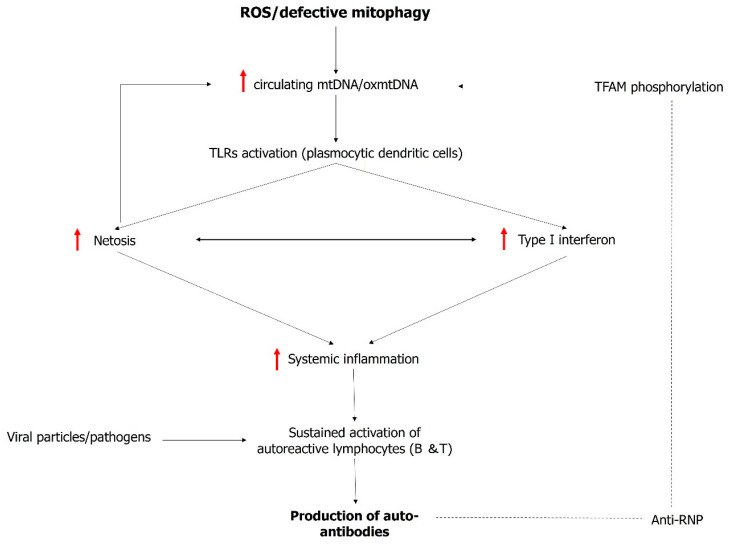
Interplay between mitochondrial dysregulation and inflammation in systemic lupus erythematosus. Red arrow = increase.

**Table 1 ijms-25-06162-t001:** Summary of the principal drugs used to treat lupus nephritis.

Standard Therapy for Lupus Nephritis			
Drugs	Mechanism of action	Indication	Comments
Cyclophosphamide with corticosteroid *	Alkylating agent: reduces number of lymphocytes (both B and T cells)	Induction phase	Two regimens: Eurolupus protocol: IV cyclophosphamide 500 mg every 2 weeks for a total of six doses (3 months). NIH protocol: IV cyclophosphamide 500–1000 mg/m^2^ once a month for 6 months.
Mycophenolate mofetil with corticosteroid *	Inhibits the synthesis of guanine nucleotides	Both induction and maintenance phase	MMF 2–3 g per day, divided into two doses (1–1.5 g twice daily) for induction phase. MMF 1–2 g per day, divided into two doses (0.5–1 g twice daily) for maintenance phase.
Belimumab with SOC	Monoclonal antibody that inhibits the BAFF pathway	Both induction and maintenance phase	Use as a corticoid-sparing agent for articular SLE. Duration of treatment: two years (***BLISS***-***LN study).***
**Non-Standard Therapy for Lupus Nephritis**			
Drugs	Mechanism of action	Indication	Comments
Rituximab	Monoclonal antibody that targets CD20 protein, which is expressed on the surface of pre-B and mature B lymphocytes	Induction phase or as rescue therapy	Could be used as monotherapy for inducing remission in pure Lupus nephritis Class V. Could be used in conjunction with low dose of MMF for induction remission in active LN (Rituxilup protocol).
Voclosporin with MMF and low dose of corticoid	Calcineurin inhibitor: inhibition of T cell activation	Both induction and maintenance phase	Duration of treatment up to three years (Aurora 2 study).
Obinutuzumab with MMF and low dose of corticoid	Monoclonal antibody that targets CD20 protein. More potent than rituximab and notably enhances antibody-dependent cellular cytotoxicity and cellular apoptosis	Induction phase	Obinutuzumab 1000 mg at day 1, then weeks 2, 24, and 26.
Tacrolimus with low dose of MMF and lose dose of corticoid	Calcineurin inhibitor: inhibition of T cell activation	Induction and maintenance phase	Part of the multitarget therapy. MMF 1–1.5 g twice daily. Tacrolimus: 2–4 mg daily, adjusted based on blood levels and clinical response.

* Considered standard of care (SOC). BAFF: B-cell activating factor, MMF: Mycophenolate mofetil, NIH: National Institutes of Health.

**Table 2 ijms-25-06162-t002:** Current mitochondrial-targeting drugs assessed in systematic lupus erythematosus.

Drug	Mechanism of Action on Mitochondria	Efficacy in SLE
Metformin	Inhibits the mitochondrial enzyme complex: mitochondrial glycerophosphate dehydrogenase	Increased renal function and reduced glomerular inflammation in a murine lupus model. No effect on lupus flare in a small, randomized control study.
Sirolimus	Restoration of mitophagy by interaction with HRES-1/Rab4 expression	Decrease in BILAG and in total dose of corticoid in patients with persistent SLE activity in a single-arm trial.
MitoQ	Mitochondrial antioxidant	Reduced NET formation, serum IFN-I, reduced immune complex deposit in kidneys in a murine model.
Idebenone	Analogue of coenzyme Q10	Decreased glomerular inflammation and fibrosis and decreased NET formation in murine model.
UMI-77	Inducer of mitophagy by interacting with the BAK/BAX pathway	Reduced glomerular inflammation, notably by decreasing T cell infiltration in the kidney in a murine model of LN.

BILAG: British Isles Lupus Assessment Group. IFN: interferon. LN: lupus nephritis.

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
