# Peer review of "Mitochondrial Dysfunction in Systemic Lupus Erythematosus with a Focus on Lupus Nephritis"

_ijms, 2024, doi:10.3390/ijms25116162_

Round 1

Reviewer 1 Report

Comments and Suggestions for Authors

The authors present a comprehensive review describing the association of mitochondrial dysfunction and lupus nephritis pathology. The text is fluent, easy to follow and the majority of the references are recent studies. However, I suggest addig 2 salient studies about mitophagy in lupus nephritis. The first one demonstrated that podocyte dysfunction in lupus nephritis is associated with reduced podocyte nestin expression and that nestin regulates nephrin through mitophagy, protecting podocytes and thus reducing proteinuria (Cell Death Dis. 2020 May 5;11(5):319. doi: 10.1038/s41419-020-2547-4.). As podocyte dysfunction and loss of nephrin is a hallmark of proteinuric diseases, this study deserves to be mentioned.

Also, it has been recently demonstrated in a salient paper that in lupus prone MRL/lpr mice the pharmacological induction of mitophagy can alleviate nephritis (Kidney Int. 2024 Apr;105(4):759-774. doi: 10.1016/j.kint.2023.12.017.). This study underscores the importance of mitophagy in disease progression as well as opens future therapeutic ways to fight lupus nephritis.

Minor issues:

The explanation of Figure 1 A,B and C panels is now within the main text, it should be moved into the figure legend instead.

The authors use the term "netosis" describing the well-known process of NET formation in SLE pathogenesis, however I suggest to write it as "NETosis".

When we discuss IFN I signature of lupus, it would be worth to decipher after first mentioning IFN I that it consists of IFN-alfa and IFN-beta.

Comments on the Quality of English Language

Minor grammar errors need to be corrected, as well as several typos in the text, such as unnecessary doubled spaces between words, or by resolving STING abbreviation the first letter should capital, etc.

Unfortunately, lines are not numbered so can not precisely point on the specific lines with the errors.

Author Response

please see the attachement

Reviewer 2 Report

Comments and Suggestions for Authors

The authors present a manuscript on mitochondrial dysfunction in systemic lupus erythematosus, with a focus on lupus nephritis.

Remarks:

1)     1) In the abstract, authors should include a clear statement of the objectives of the review, as stated at the end of the introduction.

2)     2) It is appropriate to summarize the current therapies of lupus nephritis with the use of conventional steroid and immunosuppressive drugs and officially approved and non-approved biologic drugs (anti-CD20, anti-BLyS, etc.).

3)     It is useful for the reader to add a table of mitochondrial-targeted drugs mentioned in the text, with an indication of any ongoing or completed studies on their use.
